# Airborne Bacterial and Eukaryotic Community Structure across the United Kingdom Revealed by High-Throughput Sequencing

**Hokyung Song, Ian Crawford, Jonathan R. Lloyd \*, Clare H. Robinson, Christopher Boothman, Keith Bower, Martin Gallagher \*, Grant Allen and David Topping**

Department of Earth and Environmental Sciences, The University of Manchester, Manchester M13 9PL, UK; ho-kyung.song@manchester.ac.uk (H.S.); i.crawford@manchester.ac.uk (I.C.); clare.robinson@manchester.ac.uk (C.H.R.); christopher.boothman@manchester.ac.uk (C.B.); k.bower@manchester.ac.uk (K.B.); grant.allen@manchester.ac.uk (G.A.); david.topping@manchester.ac.uk (D.T.)

\* Correspondence: jon.lloyd@manchester.ac.uk (J.R.L.); martin.gallagher@manchester.ac.uk (M.G.)

**Abstract:** Primary biological aerosols often include allergenic and pathogenic microorganisms posing potential risks to human health. Moreover, there are airborne plant and animal pathogens that may have ecological and economic impact. In this study, we used high-throughput sequencing techniques (Illumina, MiSeq) targeting the 16S rRNA genes of bacteria and the 18S rRNA genes of eukaryotes, to characterize airborne primary biological aerosols. We used a filtration system on the UK Facility for Airborne Atmospheric Measurements (FAAM) research aircraft to sample a range of primary biological aerosols across southern England overflying surface measurement sites from Chilbolton to Weybourne. We identified 30 to 60 bacterial operational taxonomic units (OTUs) and 108 to 224 eukaryotic OTUs per sample. Moreover, 16S rRNA gene sequencing identified significant numbers of genera that have not been found in atmospheric samples previously or only been described in limited number of atmospheric field studies, which are rather old or published in local journals. This includes the genera *Gordonia*, *Lautropia*, and *Psychroglaciecola*. Some of the bacterial genera found in this study include potential human pathogens, for example, *Gordonia*, *Sphingomonas*, *Chryseobacterium*, *Morganella*, *Fusobacterium*, and *Streptococcus*. 18S rRNA gene sequencing showed *Cladosporium* to be the major genus in all of the samples, which is a well-known allergen and often found in the atmosphere. There were also genetic signatures of potentially allergenic taxa; for example, *Pleosporales*, *Phoma*, and *Brassicales*. Although there was no significant clustering of bacterial and eukaryotic communities depending on the sampling location, we found meteorological factors explaining significant variations in the community composition. The findings in this study support the application of DNA-based sequencing technologies for atmospheric science studies in combination with complementary spectroscopic and microscopic techniques for improved identification of primary biological aerosols.

**Keywords:** bioaerosol; microbial community; high-throughput sequencing; health; UK

## 1. Introduction

Although the biomass levels of airborne microbial communities are considered low [1], their impact on human life is crucial. Primary biological aerosols often include allergenic and pathogenic microorganisms posing potential risks to human health [1]. According to Gupta et al. (2004), 39% of children and 30% of adults in the UK suffered from allergenic diseases, for example asthma, eczema, and hay fever [2]. Atmospheric allergens consist of fungal spores, plant pollens, dust, fragments of

plant and animals, protozoa, algae, and bacteria [3]. Due to their undesirable impacts on human health, there have been continuous efforts made to identify allergenic species in the atmosphere [3–7]. Many pathogenic species are transmitted via airborne vectors causing endemic and pandemic diseases [8]. Many prominent pandemic diseases, for example Spanish flu and tuberculosis, have been caused by bacterial or viral species transmitted through aerosols [9–12].

Moreover, there are plant and animal pathogens, which may have ecological and economic impact [13]. For example, the sporangia of *Phytophthora infestans*, which cause potato blight can survive for several hours and can travel several kilometres through the air from their place of origin [14,15]. Another example is *Mycobacterium bovis*, which causes bovine tuberculosis in livestock and wild life, impacting local ecosystems and bringing significant economic loss [16–18]. Recent reviews have highlighted the emerging threats of fungal spores to animals, plants and ecosystems including their socio-economic impacts on arable crops [19].

Microbes may also contribute to fundamental climate-ecosystem feedback processes with potential large perturbations on human-ecosystem interactions. The presence of ice nucleation active (INA) microbial species, for example, may have an indirect effect on human life through their potential role on climate change and pathogenicity to crop species. Among the INA species, *Pseudomonas syringae* has been studied extensively, and causes frost damage on plant leaves [20].

Previously, primary biological aerosols were characterized based on their morphological characteristics captured by microscopes or by culturing them and more recently ultraviolet laser-induced fluorescence (UV-LIF) methods have been developed for real-time quantification of airborne concentrations [21]. However, the application of single particle UV-LIF techniques is still under development, and while providing quantitative insight to emission mechanisms discrimination is generally limited to broad classes of biological particles rather than specific species [22]. The differences in fluorescence excitation-emission spectra are therefore not always sufficient to distinguish between different bacterial or eukaryotic taxa. However, new advances in molecular microbial ecology, offer powerful complementary approaches. For example, the advent of metagenomics and high-throughput sequencing has revolutionized microbial ecological field studies through the ability to target non-culturable microbial species, which represent the majority of microbial diversity in many natural ecosystems.

Many studies in the atmospheric science field have used high-throughput sequencing to identify primary biological aerosols. On a ground based study, for example, Li et al. (2019) [23] collected samples from Ximen in China and found correlation between urbanization level and the relative abundance of potential human pathogens. Núñez and Moreno (2019) [24] collected samples from the second tallest tower in Spain and identified large proportions of pathogens and aeroallergens in their samples. DeLeon-Rodriguez et al. (2013) [25] collected low- and high-altitude air masses in the west coast and across the USA using an aircraft and identified 17 core bacterial operational taxonomic units (OTUs) including the taxa that have potential to utilize diverse carbon components in the air. Smith et al. (2018) [26] compared bacterial community composition of the air samples collected from the ground with that of the samples collected from their flight in the USA. Maki et al. (2017) [27] collected high altitude air samples using a helicopter to compare bioaerosol composition during Asian dust periods and non-dust periods in a coastal area of Japan. Although there have been several studies which have used high-throughput sequencing to characterize airborne biological population, to the best of our knowledge, there have been no such studies to characterize airborne microbes at altitude across the UK.

In this study, we used high-throughput sequencing (Illumina, MiSeq) targeting the 16S rRNA genes of bacteria and the 18S rRNA genes of eukaryotes, to characterize airborne primary biological aerosols collected by a filtration system fitted to the UK Facility for Airborne Atmospheric Measurements (FAAM) research aircraft. We collected samples over Chilbolton, a rural area whose primary land cover comprises arable and horticultural land classes, and Weybourne, which is a coastal area located 265 km away from Chilbolton and downwind of major urban conurbations [28]. In this pilot study, we

tested the following hypotheses: (1) the airborne bacterial and eukaryotic community structure over Weybourne is different from that of Chilbolton. (2) The airborne bacterial and eukaryotic community composition will vary according to meteorological conditions under which the aircraft samples were collected which can be source apportioned.

## 2. Experiments

### 2.1. Sample Collection

Filter samples were collected using a research aircraft operated by the Facility for Airborne Atmospheric Measurements (FAAM, Cranfield, UK), UK's BAe-146-301 Atmospheric Research Aircraft. Figure 1 shows photographs of the filtration system installed in the research aircraft. The air flow-rate of the inlet system in the aircraft was 25–30 L/min. MCE (mixed cellulose ester) membrane filters with pore size of 0.45 μm and diameter of 47 mm (Millipore, Bedford, MA, USA) were used to collect bioaerosol samples for offline analysis. To sterilize the filter holders before use, they were placed in 90% isopropanol alcohol for 24 h and dried for 24 h in a fume cupboard. All operators wore nitrile gloves sterilized with 70% isopropanol alcohol wipes. The wipes were also used to clean the filter system while handling. The aircraft aerosol inlets were sterilised several times before the flight using isopropyl alcohol swabs and inspected using an endoscope. Blank handling filters (un-aspirated) were also collected to assess potential contamination.

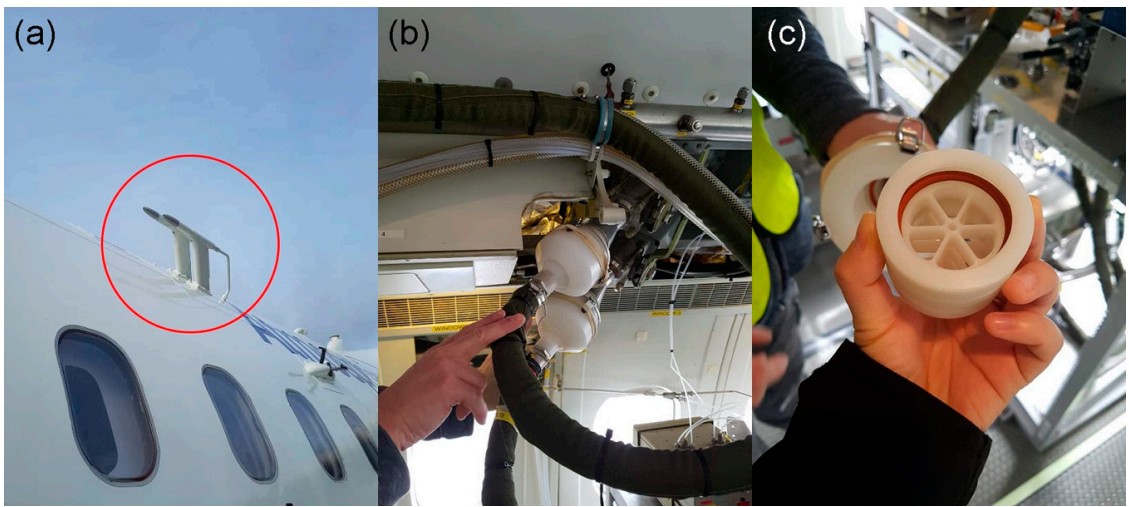

**Figure 1.** Photo of the Facility for Airborne Atmospheric Measurements (FAAM, UK) research aircraft and the filter collection system. (**a**) Aerosol inlets mounted on the aircraft fuselage. (**b**) Aerosol filter collection system attached to the inlets. (**c**) Filter holder.

Samples were collected in two locations, Chilbolton (51°8′ N, 1°26′ W) and Weybourne (52°57′ N, 1°7′ E) in the UK on 15 May 2019. Both of the sites are dominated by arable and grassland land types, with Chilbolton and Weybourne being upwind and downwind of major conurbations depending upon prevailing wind directions. We have selected these two sites to compare the in-land UK area (Chilbolton) with a typical coastal area (Weybourne). Detailed sample information including the aircraft GPS (global positioning system) location, altitude, average temperature and relative humidity can be found in Supplementary Table S1 and are archived at the FAAM aircraft database under flight reference, C173. Light surface (3–4 m s$^{-1}$) E (east) to ESE (east-southeast) winds, prevailed at Chilbolton during the flight with stronger coastal at Weybourne which was also influenced by inflow. Surface temperatures ranged from a high of 19 °C at Chilbolton to 14 °C at Weybourne with the boundary layer well mixed to a depth of 1200 m as determined by aircraft profile measurements. There was no precipitation during the flying period. Filter samples were collected at heights ranging from 430 to

440 m asl (above sea level) in the lower boundary layer, and from 600–960 m asl in the upper boundary layer (see Table S1, Supplementary Material). Filter samples were collected in parallel with various exposure times (10 min, 20 min, and 30 min typically) to help optimize the sample and analysis process and also to assess potential spatial variability over the two sites. Filter samples were also mounted in a second aircraft aerosol sampling system, but not exposed to the external air stream. These acted as blanks to assess potential contamination issues due to handling and in-cabin activities. Filters were transported directly to the laboratory in the University of Manchester and kept in a freezer before DNA extraction. The results from analysis of nine representative filter samples are presented.

## 2.2. DNA Extraction, PCR, and Sequencing

Bulk DNA samples were extracted from the filters using the DNeasy PowerWater Kit (Qiagen, Hilden, Germany) with an extraction control with an empty filter. To amplify V4 region of bacterial 16S rRNA gene, prokaryotic primers, 515F (5′-GTGYCAGCMGCCGCGGTAA-3′) and 806R (5′-GGACTACHVGGGTWTCTAAT-3′) were used [29,30]. PCR (polymerase chain reaction) condition for bacterial 16S rRNA gene was as follows: initial denaturation step at 95 °C for 2 min, 36 cycles of melting (95 °C, 30 s), annealing (58 °C, 30 s), and extension (72 °C, 2 min), and final extension at 72 °C for 5 min. To amplify V9 region of eukaryotic 18S rRNA gene, eukaryotic primers, 1391F (5′-GTACACACCGCCCGTC-3′) and EukBr (5′-TGATCCTTCTGCAGGTTCACCTAC-3′) were used [31–33]. PCR condition for eukaryotic 18S rRNA gene was as follows: initial denaturation step at 95 °C for 2 min, 37 cycles of melting (95 °C, 30 s), annealing (57 °C, 60 s), and extension (72 °C, 1.5 min), and final extension at 72 °C for 5 min. Amplified DNAs were sequenced (paired-end) based on Illumina MiSeq platform [34] following the protocol provided by manufacturer. Among the nine filter samples, only six selected samples (the samples collected over Chilbolton at the higher elevations and the samples collected over Weybourne) were sequenced for the 16S rRNA gene. Among the nine samples analysed for the 18S rRNA gene, sequencing of the "Chilbolton high 20 min" failed due to the low amount of DNA. The raw fastq formatted sequence files collected were subsequently archived in the NCBI (National Center for Biotechnology Information) SRA (sequence read archive) under project number of PRJNA629306.

## 2.3. Sequence Analysis

Paired-end sequences were combined using the PANDASeq software v. 2.8 [35]. Further sequence analysis including sequence alignment, quality control (e.g., removal of ambiguous sequences and chimeric sequences), classification, and operational taxonomic unit (OTU) clustering was performed using Mothur software v. 1.42.3 [36] following MiSeq SOP (standard operating procedure) [37]. To remove chimeric sequences, VSEARCH v. 2.13.3 [38] was used. Silva database v. 132 [39] was used for alignment and classification of sequences. OTUs were defined based on 97% sequence similarity using the OptiClust algorithm [40]. Singleton sequences, reads with sequences that are present only once in the dataset, were removed and OTUs with more than 100 reads in the extraction control were also removed. Since the Silva database provides taxonomic information only down to genus level, we used local BLAST [41] software v. 2.9.0 with the representative sequence of each OTUs against the NCBI nucleotide (nt) database [42] with e-value cutoff of $10^{-10}$.

## 2.4. Statistical Analysis

For diversity analysis, samples were sub-sampled into 15,901 reads per sample for prokaryotes and 13,509 reads per sample for eukaryotes. To draw an nMDS (non-metric multidimensional scaling) plot, the number of reads were square root transformed and the Bray–Curtis dissimilarity was calculated. To test the significance of community distances between sampling location, an analysis of similarity (ANOSIM) test was performed. nMDS plot visualization and ANOSIM test was performed using Primer v. 6 [43]. To understand the relationship between meteorological variables (average temperature and relative humidity during the sampling time, Table S1) and prokaryotic (bacterial) and eukaryotic

community structures, we performed canonical correspondence analysis (CCA) and redundancy analysis (RDA) with square root transformed community data. To test the significance of constraints (meteorological factors), permutation tests were performed with 999 permutations and significant meteorological factors were forward selected. CCA were performed instead of RDA when the response data have a gradient larger than four SD units. CCA and RDA were performed using Canoco v. 5 [44].

## 3. Results and Discussion

Supplementary Figure S1 shows a series of 96-h Lagrangian back trajectories for air masses arriving over Chilbolton and Weybourne at altitudes between 380 m and 950 m, calculated using the Hybrid Single Particle Lagrangian Integrated Trajectory (HYSPLIT) modelling system with full three-dimensional (3-D) advection [45,46]. HYSPLIT was driven using Global Data Assimilation System (GDAS) 0.5 degree gridded meteorological reanalysis data. The trajectories illustrate a prevailing easterly flow near to the surface (below 2000 m) in a descending air mass characterized by high pressure. This is consistent with an interpretation that only local land-based bioaerosol sources over South East England would have had dynamic contact with the surface environment, for a period of 6 to 12 h prior to sampling over Chilbolton and Weybourne.

### 3.1. Bacterial Community Structure and Diversity

Figure 2 shows the bacterial phylum compositions of the samples. The most abundant phylum was Proteobacteria (averaged relative abundance of 45.5%), Actinobacteria (28.7%), Bacteroidetes (11.1%), and Firmicutes (6.7%). This corresponds to earlier studies on the airborne microbial community. For example, Núñez and Moreno (2020) [24] found Proteobacteria, and Actinobacteria being the most abundant phyla in their samples collected through an air sampler. Smith et al. (2018) [26] found Firmicutes, Proteobacteria, Actinobacteria, and Bacteroidetes, being the most abundant ones in their flight samples, although they were also abundant in their hardware samples. Maki et al. (2017) [27] found high relative abundance of Actinobacteria and Proteobacteria during non-dust periods at high altitudes over the Noto peninsula in Japan.

The bacterial genus composition shows only a few genera overlapping between samples indicating a large heterogeneity across the samples (Figure 3). The most abundant genus in each of the samples collected in Chilbolton at high elevation (range of 950–1000 m asl) in the time durations of 10 min, 20 min, and 30 min was *Gordonia* (64.8%), *Fusobacterium* (22.8%), and *Candidatus Udaeobacter* (14.0%) respectively. The most abundant genus (except for those only classified down to family level) in each of the samples collected in Weybourne (height range of 650–1000 m) in the time duration of 10 min, 20 min, and 30 min was *Lautropia* (39.5%), *Morganella* (38.8%), and *Psychroglaciecola* (31.7%) respectively. Interestingly, there were significant numbers of genera that have not been found (or found only in few cases) in atmospheric samples previously, for example, *Gordonia*, *Psychroglaciecola*, *Lautropia*, *Morganella*, and *Fusobacterium*. Some of the bacterial genera we found include potential human pathogens, for example, *Gordonia*, *Morganella*, *Fusobacterium*, and *Sphingomonas.*

*Gordonia* species have been isolated from soil, sludge, and clinical samples [47], with several species reported to be opportunistic human pathogens causing secondary infections, such as septicaemia, cutaneous infection, and pulmonary infections, in immunocompromised and immunosuppressive individuals [47–52]. The closest hit from the blast search against OTU008, the only OUT belonging to *Gordonia* in our samples, was *Gordonia iterans* (Supplementary Table S2), which has been isolated from a patient with pneumonia [53] posing potential risk to human health.

The presence of the DNA most closely related to *Fusobacterium* species, known to cause periodontal disease in humans [54] is interesting, as representatives of this genus are considered obligate anaerobes [55]. This would seem to make their presence in an active form unlikely in an aerosol sample. However, they are known to form biofilms, which can help them persist in aerated environments [56], consistent with more recent studies showing the presence of bacterial biofilms in aerosols [57].



OTU013, the only OTU affiliated with *Morganella* in our samples, had 100% similarity with *Morganella morganii* (Supplementary Table S2), which is also a common opportunistic pathogen found in natural environments such as soil and water, and also in healthcare facilities [58]. They are often associated with diverse diseases including urinary tract infection, sepsis and pneumonia [59]. However, so far their transmission mode has not been fully understood [59,60]. The presence of their DNA in our filter sample suggests that aerosols could be a potential transmission medium of these bacterial species.

There were also potential ice-nucleating bacterial species including *Pseudomonas* (1.09%) and *Xanthomonas unclassified* (0.03%) detected in our samples but with low relative abundance on this day. These bacterial genera have been commonly found in rain and snow samples and also in other atmospheric samples [61]. They have been studied extensively due to their adverse effect on agricultural systems.

Figure 4 shows an nMDS plot generated based on Bray–Curtis distance of bacterial communities between samples. In contrast to our hypothesis, there was no sign of clustering by sampling location (ANOSIM, $p = 0.7$). However, the CCA analysis with forward selection of environmental variables showed temperature as a significant explanatory variable for community composition ($p = 0.041$, Figure S2). The links between temperature with airborne bacterial species have also been elucidated in many other studies [62–64]. For example, Bertolini et al. (2013) [62] collected 10 samples in Italy in different seasons and found that the airborne microbial communities were mainly separated by average daily temperatures. However, considering the very narrow range of temperature in our samples collected for bacterial gene sequencing (7.5–8.5 °C, Table S1) there should be further sampling efforts to cover a wider range of temperature to have further discussions on this result.

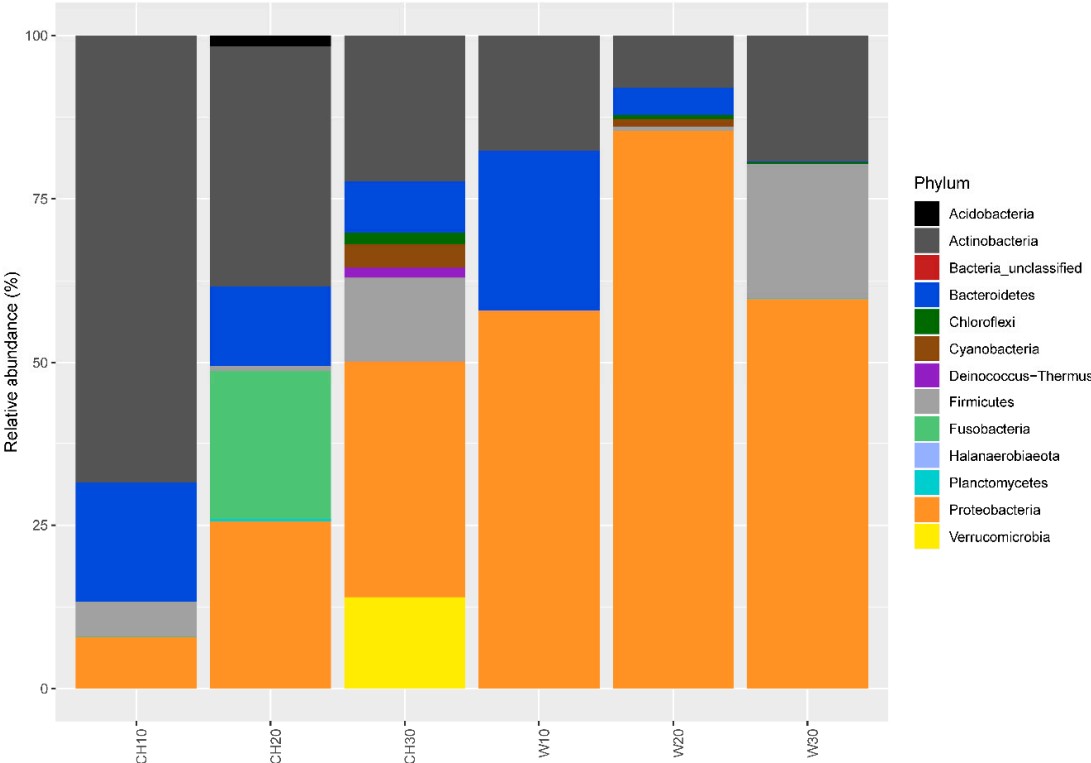

**Figure 2.** Bacterial phylum composition of the samples. Each of CH10, CH20, CH30, indicates samples collected in Chilbolton at high elevation (range of 950–1000 m asl) in the time duration of 10, 20 and 30 min. Each of W10, W20, W30, indicates samples collected over Weybourne (within a height range of 650–1000 m) with exposure durations of 10, 20, and 30 min., respectively.

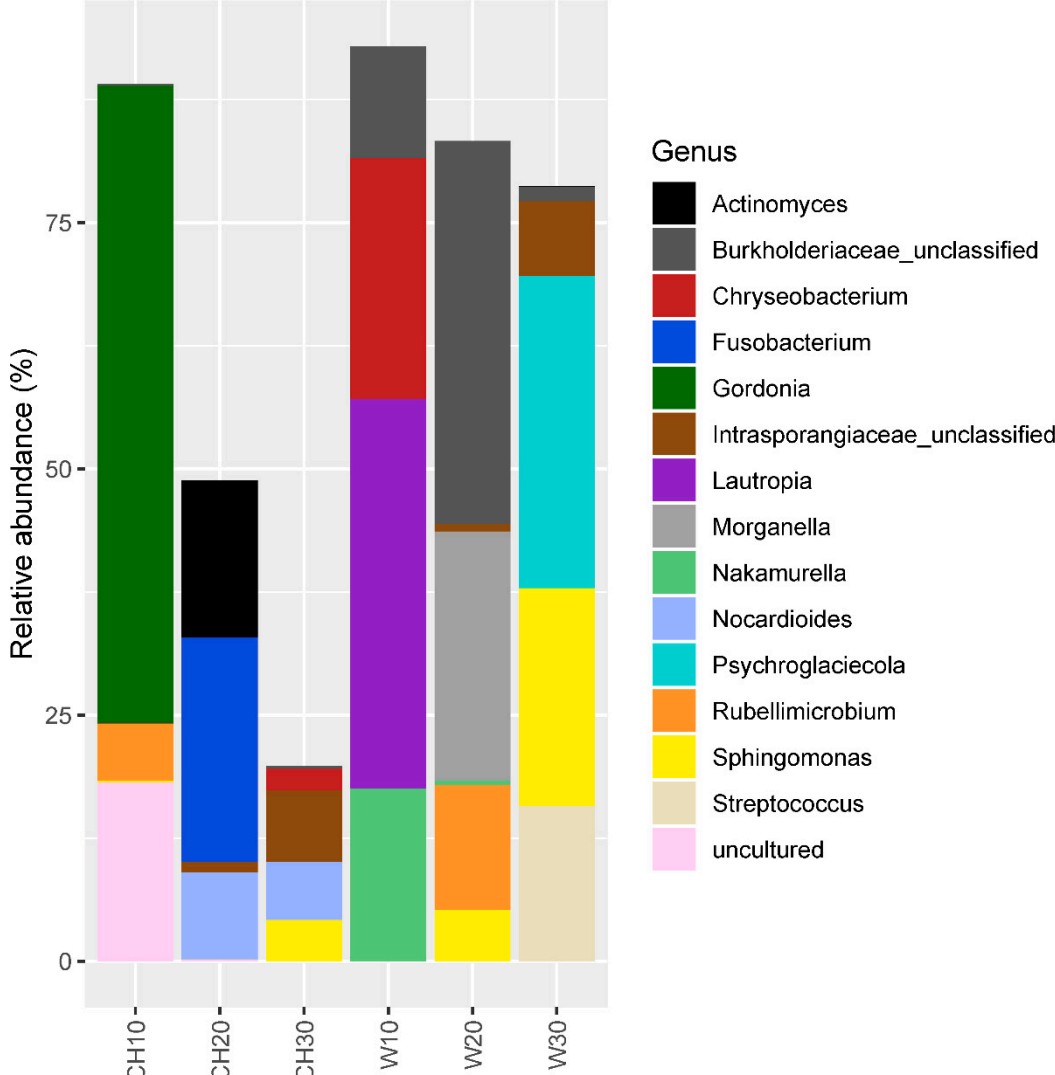

**Figure 3.** The 15 most abundant bacterial genera found the samples. Each of CH10, CH20, CH30, indicates samples collected in Chilbolton at high elevation (range of 950–1000 m asl) in the time duration of 10 min, 20 min, and 30 min. Each of W10, W20, W30, indicates samples collected in Weybourne (height range of 650–1000 m) in the time duration of 10 min, 20 min, and 30 min.

Regarding diversity, there were 30 to 60 bacterial OTUs identified per sample (Figure S3). There was no significant correlation between the bacterial Shannon diversity and temperature (rho = 0.6, $p = 0.24$) nor humidity (rho = 0.37, $p = 0.50$). There was indication of an increase in the bacterial diversity with increase in sampling time duration at each site, although the results were not statistically significant (Spearman's rho = 1, $p = 0.33$ in both Chilbolton and Weybourne) due to the limited number of samples.

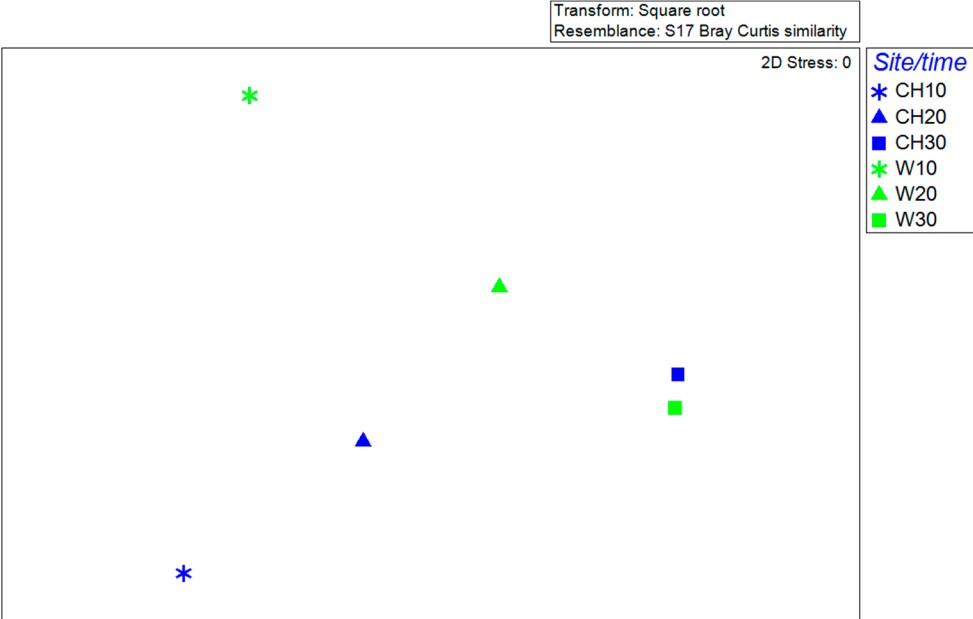

**Figure 4.** Non-metric multidimensional (nMDS) plot of bacterial communities. Each of CH10, CH20, CH30, indicates samples collected in Chilbolton at high elevation (range of 950–1000 m asl) in the time duration of 10 min, 20 min, and 30 min. Each of W10, W20, W30, indicates samples collected in Weybourne (height range of 650–1000 m) in the time duration of 10 min, 20 min, and 30 min.

### 3.2. Eukaryotic Community Structure and Diversity

Figure 5 shows the phylum level composition of eukaryotic community in each of the sample. The most abundant phylum was Ascomycota (averaged relative abundance of 39.6%) followed by Phragmoplastophyta (37.4%), and Basidiomycota (8.7%). Ascomycota and Basidiomycota are the two largest fungal phyla, which are commonly found in many environments including the atmosphere. Phragmoplastophyta consists of several clades of green algae and land plants. As the majority of the eukaryotic composition in the air are fungal spores, fungal hyphal fragments, plant pollens, or algae [65], the major phylum composition observed is not surprising.

Figure 6 shows the genus level composition of the eukaryotic communities. Unlike the bacterial communities, there were overlapping taxa between the samples. *Cladosporium* was one of the major genera detected in all of the samples (average relative abundance of 14.3%), which is a well-known allergen and often reported in the atmosphere [6,66,67]. Due to the small size of their conidia with a branched chain shape, *Cladosporium* species can travel over long distances by winds [67]. There are a large number of *Cladosporium* species that are pathogenic to plants, and rare reports of pathogenicity to humans [68,69].

*Phoma*, which was also one of the major genera detected (found in four samples, average relative abundance of 5.5%), includes a large number of phytopathogenic species causing major impacts on several economically important crops [70]. They are also one of the allergenic species that are commonly found in atmospheric samples [6]. Although very rare, there are several reports that show pathogenicity of *Phoma* species to immunosuppressed patients [71].

There were also genetic signatures of other potentially allergenic taxa found in our samples for example, species of *Brassicales*, *Ustilaginaceae*, and *Pleosporales* [5,6,72]. Moreover, a large number of other potential plant pathogens were identified, for example, the unclassified genera belonging to the family *Pleosporales* and *Ustilaginaceae* and the genera *Blumeria*, *Ustilaginaceae*, *Sclerotinia*, *Delphinella*, and *Itersonilia* [73–79]. Although it was not one of the major taxa detected, *Phytophthora* was also found in three samples (average relative abundance of 1.03%). *Phytophthora* species are well-known plant pathogens, which cause potato blight.

Figure 7 shows an nMDS plot generated based on Bray–Curtis distance of eukaryotic communities between samples. In contrast to our expectation, there was no significance difference in eukaryotic composition between the two sampling sites (ANOSIM, $p = 0.14$). The eukaryotic composition was instead significantly affected by relative humidity ($p = 0.008$) based on the result of the RDA (Figure S4). Although there have been many studies on the seasonal effect on the plant pollen and fungal spore abundance and composition in air [80–83], we are aware of no study, which shows an effect of relative humidity on eukaryotic community composition within a short sampling period as in our study. As the plant community and the fungal communities at the surface will not have changed significantly during the experimental period, the impact of humidity would be more related to the differences in the dispersal of various plant and fungal species. As the shape and size of plant pollens and fungal spores differ by different taxa, the dispersal of plant pollens could be dependent on the humidity of the air.

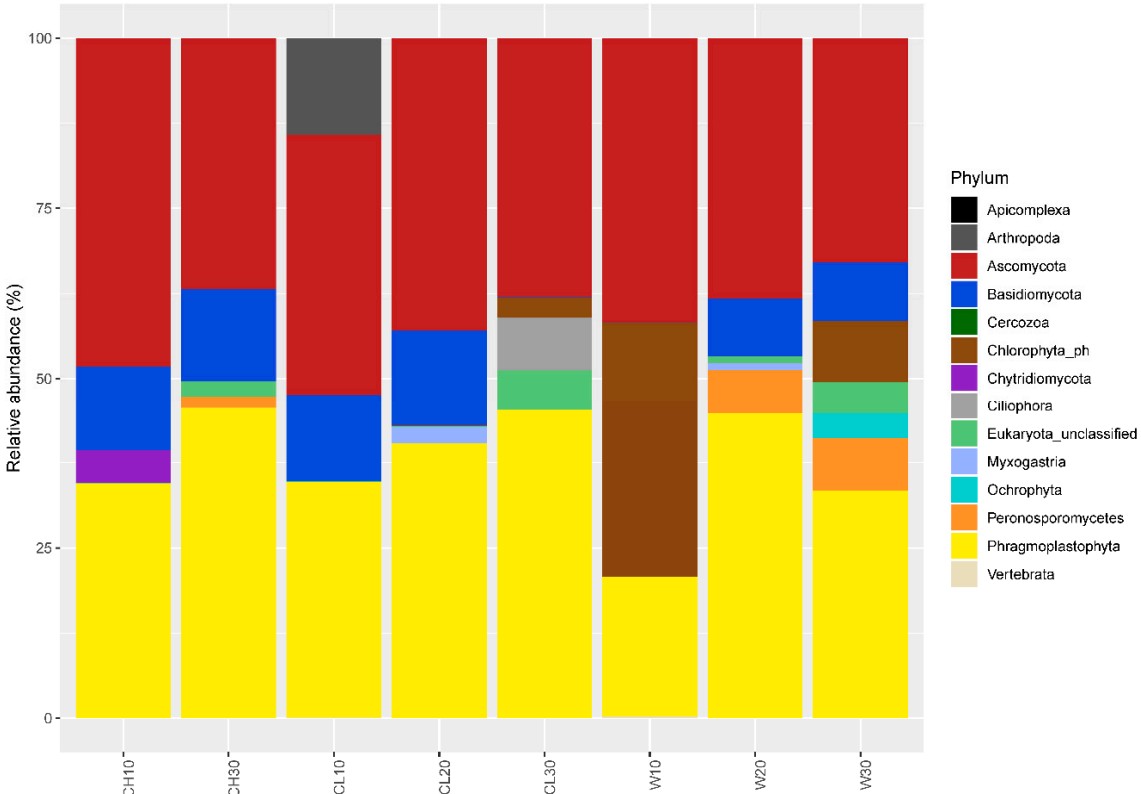

**Figure 5.** Eukaryotic phylum composition of the samples. Each of CH10, CH30 indicates samples collected in Chilbolton at high elevation (range of 950–1000 m asl) in the time duration of 10 min and 30 min. Each of CL10, CL20, CL30, indicates samples collected in Chilbolton at low elevation (range of 300–450 m asl) in the time duration of 10, 20, and 30 min. W10, W20, W30, indicates the different samples collected in Weybourne (height range of 650–1000 m) with time durations of 10, 20, and 30 min respectively.

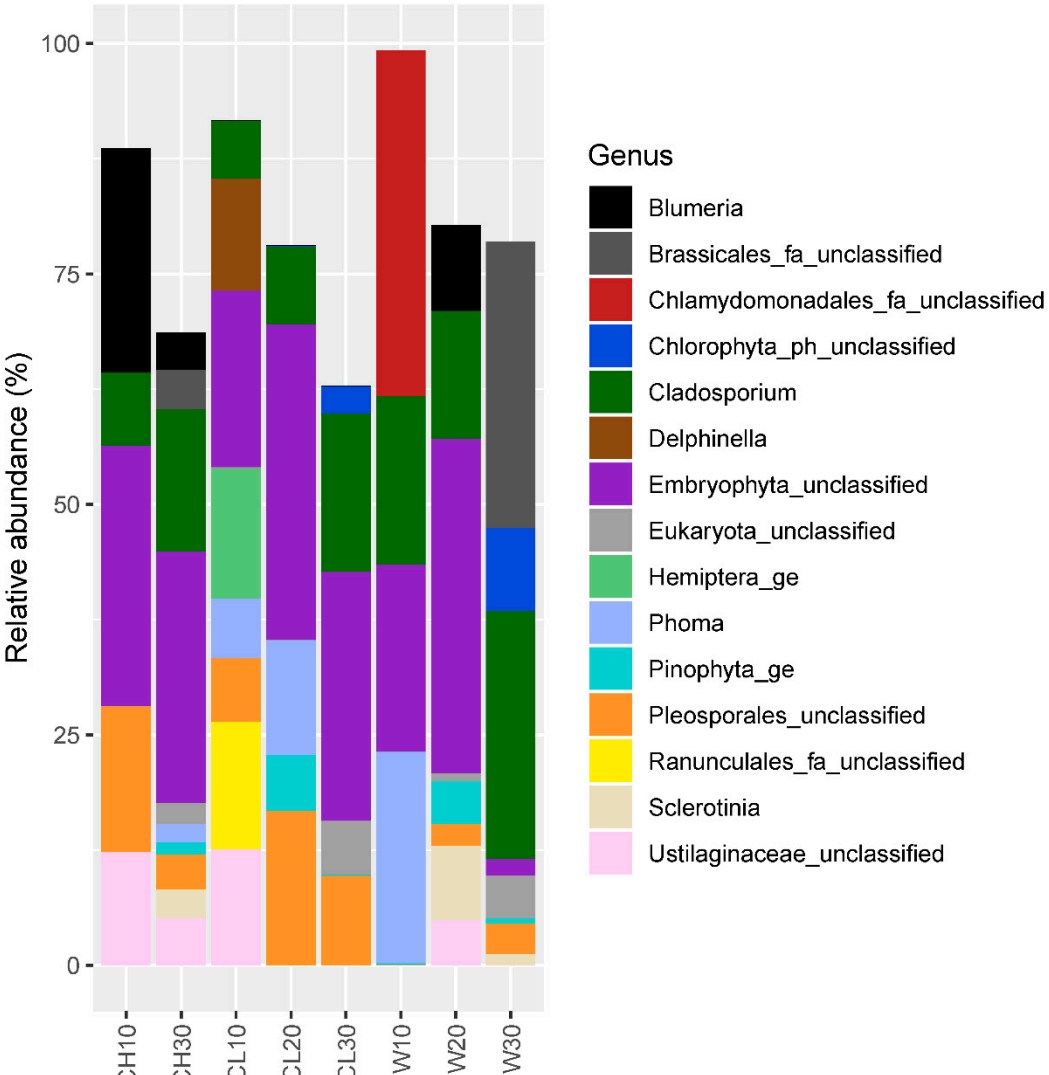

**Figure 6.** The 15 most abundant eukaryotic genera detected in the samples. Labels CH10 and CH30 indicate the samples collected over Chilbolton at high elevation (ranging from 950–1000 m asl) and with exposure times of 10 and 30 min. CL10, CL20, and CL30 indicate samples collected over Chilbolton at low elevation (ranging from 300–450 m asl) with exposure times of 10, 20, and 30 min respectively. W10, W20, W30 indicates the samples collected over Weybourne (height range of 650–1000 m) with exposure times of 10, 20, and 30 min respectively.

In terms of diversity, there were 108 to 224 eukaryotic OTUs identified per sample (Figure S5). The eukaryotic Shannon diversity did not show any significant correlation with temperature (rho = 0.33, $p$ = 0.43) nor humidity (rho = 0.31, $p$ = 0.46). There was also no signal of increase in the Shannon diversity in relation to sampling time duration.

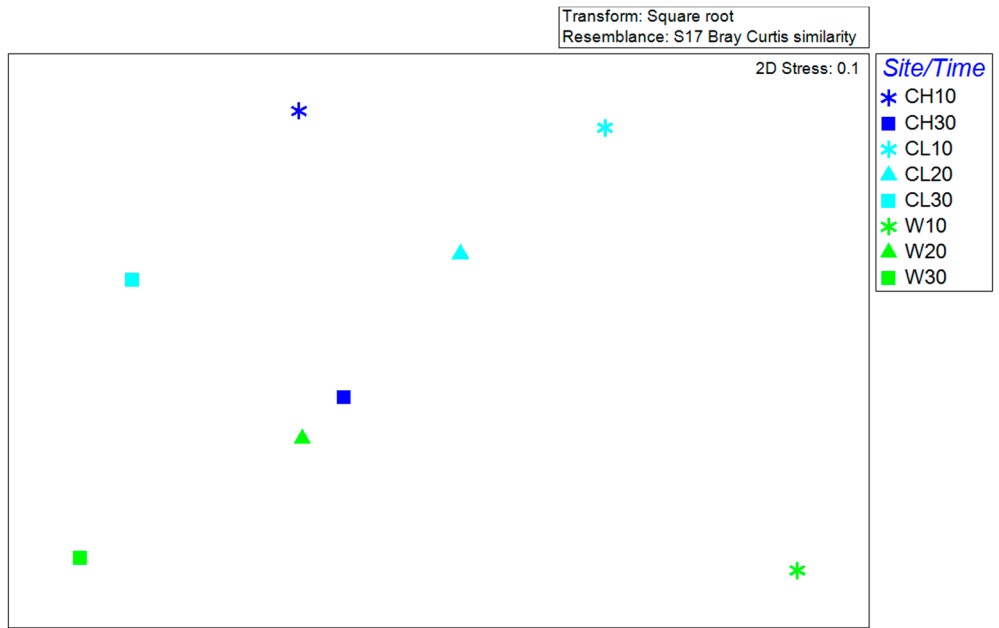

**Figure 7.** Non-metric multidimensional (nMDS) plot of eukaryotic communities. Each of CH10, CH30 indicates samples collected in Chilbolton at high elevation (range of 950–1000 m asl) in the time duration of 10 min and 30 min. Each of CL10, CL20, CL30, indicates samples collected in Chilbolton at low elevation (range of 300–450 m asl) in the time duration of 10 min, 20 min, and 30 min. Each of W10, W20, W30, indicates samples collected in Weybourne (height range of 650–1000 m) in the time duration of 10 min, 20 min, and 30 min.

### 3.3. Possible Limitation of the Study

First, although we have excluded the OTUs that were abundant (more than 100 reads) in the extraction control, it should be noted that some of the genera found in our samples are known contaminants of commercial DNA extraction kits [84,85]. Second, for eukaryotic sequences, it was difficult to identify taxonomy down to genus level (Figure 6, Supplementary Table S3). To improve resolution of taxonomic assignments of fungal taxa we found in our filter samples, it will be necessary to sequence for internal transcribed spacers (ITS) region of fungal DNA. Last, the number of samples used in this study totalled only six for bacterial community and eight for eukaryotic community analyses, which is clearly limited and makes reaching a general conclusion challenging, especially given the wide variation between community compositions noted. Further sampling efforts are necessary for a comprehensive understanding of primary biological composition over the study sites.

The findings in this study support the application of DNA-based sequencing technologies in atmospheric science studies in combination with complementary spectroscopic techniques and microscopy for improved identification of primary biological aerosols. Absolute quantification of bacterial and eukaryotic taxa and taxonomic groups of interest (e.g., ice-nucleating bacteria, fungi), should also useful. Functional gene analysis using Q-PCR and whole genome sequencing, with complementary studies on proteomics will be the next step forward to understand better the functionality of airborne bacterial and eukaryotic communities. The study also demonstrates for the first time the use of the FAAM research aircraft to conduct airborne microbiological sampling to support aerobiology applications.

## 4. Conclusions

Using the DNA-based sequencing techniques, we were able to identify diverse taxonomic groups in the airborne samples representative of Southern UK. Bacterial and eukaryotic taxa that have potential pathogenicity to human were identified which need to be studied more in a detail in relation to public

health. In addition, plant pathogens were identified, which could play a role as an early warning to disease spread to agricultural systems and ultimately impact crop yields. Very few taxonomic groups, which have potential ice-nucleation activity (and might affect local and global climate) were found, suggesting constraints on biological aerosol mediated ice activation in forecast models. It is surprising that there were no significant differences in the bacterial and eukaryotic communities between the sampling locations. Whilst this may be due to the low number of samples, the similarity confirms the consistency of air mass trajectories sampled over the two locations. The significant influence of the meteorological conditions on the relative community composition however invites further investigation on differential influence of temperatures and humidity on the aerobiome and dispersal of genera found in this study.

**Supplementary Materials:** The following are available online at http://www.mdpi.com/2073-4433/11/8/802/s1, Figure S1: 96 h HYSPLIT Lagrangian back trajectories of the sampled air, Figure S2: Canonical correspondence analysis result plot. Figure S3: Number of bacterial OTUs. Figure S4: Redundancy analysis result plot. Figure S5: Number of Eukaryotic OTUs, Table S1: Detailed information of collected filter samples. Table S2: Blast result of the 20 most abundant bacterial OTUs. Table S3: Blast result of the 20 most abundant eukaryotic OTUs.

**Author Contributions:** Conceptualization, J.R.L., C.H.R., M.G., G.A., and D.T.; Methodology, M.G., I.C., C.B., K.B.; Formal Analysis, H.S.; Investigation, I.C., C.B., K.B., M.G.; Writing—Original Draft Preparation, H.S.; Writing—Review & Editing, I.C., J.R.L., C.H.R., M.G., and G.A.; Visualization, H.S.; Supervision, J.R.L. and M.G.; Project Administration, M.G., K.B.; Funding Acquisition, M.G. All authors have read and agreed to the published version of the manuscript.

**Funding:** The authors acknowledge the support of NERC via grant NE/S002049/1.

**Acknowledgments:** Airborne data were obtained using the BAe-146-301 Atmospheric Research Aircraft [ARA] flown by Airtask Ltd., and managed by the FAAM Airborne Laboratory, jointly operated by UKRI and the National Centre for Atmospheric Science. The staff of FAAM and Airtask Ltd., and in particular their pilots, are thanked for their dedication in making the flight a success. Hokyung Song and Ian Crawford were funded by the NERC grant BIOARC, NE/S002049/1. The authors gratefully acknowledge the NOAA Air Resources Laboratory (ARL) for the provision of the HYSPLIT transport and dispersion model and READY website (https://www.ready.noaa.gov) used in this publication.

**Conflicts of Interest:** The authors declare no conflict of interest.

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
