# Peer review of "Airborne Bacterial and Eukaryotic Community Structure across the United Kingdom Revealed by High-Throughput Sequencing"

_atmosphere, doi:10.3390/atmos11080802_

Round 1

Reviewer 1 Report

In this manuscript, the authors have reported the results on a metagenomic analysis of airborne bacteria and fungi found in the atmosphere of southern England. For air collecting, they used a filtration system on the UK FAAM research aircraft that was a novel approach to conduct airborne microbiological sampling. In this way, the authors could assess the composition of microbial aerosols in the atmosphere. The study has well been designed, conducted and described. However, some improvements in the text might be considered by the authors.

  1. The authors should plan clear aims of the study, showing assumptions that are novel and worthy verification by analysis of atmospheric air. The statement that they wanted to characterize airborne primary biological aerosols is scientifically poor. Besides, information on the results of other researches using the NGS for studies of atmospheric bioaerosols might precede the aims. Their report presented in the manuscript is not the first in the study area. Thus, their work might relate to other results showing what they have wanted to add to the problem.
  2. They should explain in the text the reason of air analysing in two places of southern England (l. 93). Were there unique conditions affecting the microbiome compositions?
  3. The cited reference page cannot be found by a reader (l. 120). Thus, this citing is useless.
  4. Names of species and genera might be in italics for every title of referenced articles.
  5. The aims and conclusions might correspond to each other.

Reviewer 2 Report

General comments

   This research reported on the microbial community structures suspended at high altitude over the United Kingdom (UK) areas. The authors revealed that several kinds of bacteria and fungi are transported at the atmosphere at the two locations of UK. The bacterial communities are found to contain some dominant species which have not detected at high altitudes previously. I think  this paper is the first reports investigating the airborne microbes at high altitudes of UK, so this paper may include informative data which can be compared with other sampling data of Asia, USA or other European areas (Hot spot areas for bioaerosol research). However, there are some issues that are related to the manuscript structures as a research articles. Moreover, I recommend the authors discuss about the difference between sampling locations and among sampling periods.

    I think this manuscript can be published after major revision.

Some major comments:

  1. The relative abundances of microbial categories (the genus levels in this paper) have to be explained before discussing about the dominant microbial categories. So I think the discussion about the microbial structures is vague in this manuscript.

  1. Although there are difference of microbial community structure between two locations, the influences of location variation (or sampling date) have to be discussed. Addition to trajectory analysis, more environmental factors, such as humidity, temperatures, precipitation, particle concentrations and so on, can support this discussion. If possible, these factors have to inserted by using the public data of meteorological survey centers.

  1. I think this paper focused on 1) pathogens in airborne microbes, 2) metagenome analyses using high-throughput DNA sequencing and 3) high altitude sampling in UK area (European island). From the view of these focus points, the sections of Introduction and Result & Discussion should be redrafted. Detail decisions are described at the minor comments.

Some minor comments:

L13: The term “next generation sequencing” should be changed to “high throughput DNA sequencing”, because MiSeq is not next generation no longer.

L16: Please define FAAM.

L18: The term “eukaryotic” is changed to “fungal” better, because fugal communities are discussed here mainly. If changing, all parts “eukaryotic” or  “eukaryotie” of this manuscript are needed to be revised.

L21-22: The parts “For example~” is fragmented.

L50-59: I think INA is not important for this paper. The species of IN microorganisms have not be determined completely, so the IN assay using isolates is needed for discussing about microbial INA.

L60-71: The contribution of metagenome analyses for airborne microbes should be introduced using more references. Additionally, the previous reports relating to the airborne microbial sampling at high altitudes are also needed to be introduced here.

L71-74: Please redraft the aim of this study to be more clear.

L93: longitude and latitude of sampling sites are needed.

L147-155: More environmental conditions during sampling periods have to be described more and then the authors can discuss about differences between sampling sites or among sampling periods.

L156-205: As described, the relative abundances of the sequences at genus levels are needed before discussing about dominant microorganisms. Moreover, almost discussion are speculation in dependence on bacterial species detected from DNA database. So please shorten this part.

L163-164: The term “the genera” is needed on the front of genera manes.

Fig.2: Why are CH30 and W30 closed? This reason has to be inserted to discussion. The comparison among sampling periods and locations should be explained more and discussed about the comparison results.

L206-247: This section is also redrafted as the similar to bacterial section.

Fig.4: Why are CH30 and W20 closed? I think CL can be form a classtor  that is different from W clusstor. Please discuss about the difference between the sampling sites.

L248-258:  This parts can be moved to Conclusion section. Instead of this, please compare the diversities between bacteria and fungi  here and discuss about this.

Round 2

Reviewer 2 Report

The authors were attempted to revise the manuscript depending on reviewers’ comments. The revised version has been improved to be clearer and sophisticated.

 I recommend this paper is accepted in this journal after minor revision.

Why did the sample WH10 show high abundances of Actinobacteria?

Other references relating to high altitude surveys over island area should be also cited. For example, the following reference would support and supplement the discussion of the authors’ paper.
